# Determinants of cervical high-risk human papillomavirus positivity among Rwandan women living with human immunodeficiency virus

Gad Murenzi[1*], Faustin Kanyabwisha[1], Maria Demarco[2], Benjamin Muhoza[1],
Kristen Hansen[2], Jean Paul Mivumbi[1], Anthere Murangwa[3], Theogene Rurangwa[3],
Thierry Muvunyi Zawadi[4], Gallican Kubwimana[1], Julia C. Gage[2], Tiffany Hébert[5],
Adebola Adedimeji[6], Laetitia Nyirazinyoye[7], Marcel Yotebieng[5], Leon Mutesa[8],
Kathryn Anastos[5], Philip E. Castle[2]

1 Einstein-Rwanda Research and Capacity Building Program, Research for Development and Rwanda Military Referral and Teaching Hospital, Kigali, Rwanda, 2 National Cancer Institute, National Institutes of Health, Bethesda, Maryland, United States of America, 3 Rwanda Military Referral and Teaching Hospital, Kigali, Rwanda, 4 King Faisal Hospital, Kigali, Rwanda, 5 Albert Einstein College of Medicine, Bronx, New York, United States of America, 6 Wake Forest University School of Medicine, Winston-Salem, North Carolina, United States of America, 7 School of Public Health, College of Medicine and Health Sciences, University of Rwanda, Kigali, Rwanda, 8 Center for Human Genetics, College of Medicine and Health Sciences, University of Rwanda, Kigali, Rwanda

* gadcollins@gmail.com

## Abstract

### Introduction

There are few data on the prevalence of cervical high-risk human papillomavirus (hrHPV), the necessary cause of cervical cancer, and its determinants among Rwandan women living with human immunodeficiency virus (HIV). We therefore aimed to assess the determinants of hrHPV positivity among Rwandan women living with HIV (WLWH).

### Methods

We conducted a cervical cancer screening study of ~5,000 WLWH aged 30–54 years and living in Kigali, Rwanda, but originally from all provinces in the country, from 2016–2018. Women were tested for hrHPV by the Xpert assay (Cepheid, Sunnyvale, CA, USA). A nurse-administered questionnaire collected data on demographics and HPV/cervical cancer risk factors. Women without evidence of cervical precancer and cancer were included in the analysis.

### Results

Women included in this analysis (N = 4,880) had a mean age of 40 years, > 98% were on antiretroviral therapy, and 61% had a CD4 count of ≥500 cells/µL. High-risk

**Data availability statement:** Data cannot be shared publicly because of restrictions about data sharing by the Rwanda National Ethics Committee and the Government of Rwanda. RNEC contact for further details: ndahindwa@gmail.com and/or valentineingabire6@gmail.com.

**Funding:** KA, LM, MY and AA (U54CA254568) KA and MY (U01AI096299) Funder: National Institutes of Health The funder did not play any role in the study design, data collection and analysis, decision to publish, or preparation of the manuscript.

**Competing interests:** The authors have no competing interests. However, in the past 3 years, Dr. Castle (PEC) has received HPV tests and assays for research at a reduced or no cost from Cepheid and Atila Biosystems. Additionally, Cepheid provided Xpert HPV tests at reduced cost and loaned a GeneXpert system/machine for this study.

HPV prevalence was 25.5% [95% confidence interval (CI)=24.3%−26.8%] and the prevalence decreased with older ages and higher CD4 counts ($p_{trend}$<0.001 for both). High-risk HPV prevalence was higher for those who reported their first sex before 16 years, had their first child before 18 years, had more sexual partners over their lifetime and in the last six months, and those with lower CD4 cell count ($p_{trend}$<0.001 for all). A CD4 count of <200 (vs. >500) per µL was most strongly associated with being hrHPV positive (adjusted odds ratio=2.7, 95%CI=2.1–3.6).

## Conclusions

Our findings highlight the role of CD4 counts, as a measure of HIV control and immunity, in controlling hrHPV infection, which could potentially impact cervical cancer control among this high-risk population. Raising awareness on various associated factors coupled with integrating HPV and cervical cancer awareness in HIV care could help control this double burden of disease.

## Introduction

High-risk human papillomavirus (hrHPV) causes almost all cervical cancer [1], the fourth most common cause of cancer morbidity and mortality among women globally [2]. Cervical cancer, an AIDS-defining malignancy [3], is the most common cancer and cause of cancer deaths [2] in most of sub-Saharan Africa (SSA), which also carries the heaviest HIV burden [4].

The 2018 World Health Organization's (WHO) strategy to accelerate the elimination of cervical cancer [5] as a public health problem (≤4 cases per 100,000 women) recommends screening 70% of eligible women (at age 35 and again at 45 years) using a high-performance test. The currently recognized high-performance test is HPV DNA testing, which is yet to be rolled out and accessible in many SSA countries, hence negatively impacting achieving the 90-70-90 targets laid out in the WHO elimination strategy. That low availability of routine HPV DNA testing also means that there are limited data on the prevalence of hrHPV and its determinants among women living with HIV (WLWH) in SSA.

The second edition of the WHO guidelines for the screening and treatment of cervical precancerous lesions for cervical cancer prevention [6] identify WLWH to be at higher risk for cervical cancer with recommendations to begin their screening at a younger age (25 years) compared to their HIV-negative counterparts (at 30 years) and more frequently, every three years versus every five years for HIV-negative women. These recommendations highlight the need to gather more evidence about hrHPV and its determinants among WLWH.

In 2019, Rwanda launched a cervical cancer screening program with a shift to HPV DNA testing and following the WHO guidelines with risk-based screening (HIV status). This program, which was first piloted in a few districts for feasibility assessment, achieved some milestones but for it to be sustainable, more investment in HPV testing is required and this requires evidence for policy formulation and

evidence-based resource allocation. Therefore, gathering evidence on the burden of hrHPV and its determinants among Rwandan women, especially WLWH given their increased risk, is essential for informing cervical cancer screening strategies in Rwanda.

Although hrHPV is known to be a necessary but not sufficient causal factor for cervical cancer, other factors have been associated with cervical cancer and with hrHPV itself implying that the development of cervical cancer is multifactorial. Multiple factors, including age, number of sex partners, parity, age at first sex, smoking, oral contraceptive use and CD4 cell count for WLWH, have been studied as determinants for hrHPV positivity. There is conflicting evidence with various studies showing an association [7–10] of some of those factors with hrHPV positivity and others showing no association [11,12]. We therefore measured hrHPV prevalence and its determinants among screening-age Rwandan WLWH.

## Materials and methods

### Study design, population and setting

We conducted a cross-sectional study of ~5,000 WLWH aged 30–54 years and living in Kigali, Rwanda, but originating from all provinces in the country, who were screened for cervical cancer between March 22, 2016 and August 9, 2018. WLWH were enrolled from partner HIV clinics to achieve the calculated sample size which would allow sufficient power to compare endpoints. The study protocol, including how and where women were recruited and enrolled, has been previously described in detail [13]. In brief, WLWH were screened for cervical cancer using HPV-DNA testing and visual inspection with acetic acid (VIA). Participants with a positive screening test [positive for hrHPV by the Xpert HPV test (Cepheid, Sunnyvale, CA, USA) and/or by VIA], were referred for rigorous colposcopic evaluation, including a 4-quadrant microbiopsy/biopsy protocol [14] and specimen collection for biomarker testing. The study protocol was reviewed and approved annually by the Rwanda National Ethics Committee with reference numbers: 217/RNEC/2015, 878/RNEC/2016, 317/RNEC/2017 and 720/RNEC/2018. All participants provided written informed consent.

### Data collection methods

A nurse-administered questionnaire collected data on demographics and HPV/cervical cancer risk factors, and specimens were collected in PreservCyt (Hologic, Bedford, MA, USA) for HPV DNA testing using the Xpert assay. Screen-positive women underwent colposcopy and a four-quadrant biopsy protocol [14] with treatment of biopsy-confirmed disease using thermal ablation (TA) and large loop excision of the transformation zone (LLETZ) according to eligibility. Some women with lesions not eligible for TA or LLETZ and those suspicious for cancer were referred to a gynecologist for further management. HIV data including CD4 cell count, viral load results, and ART data were extracted from the patient electronic medical records (OpenMRS) as indicated in the study protocol.

### Laboratory testing

HPV DNA testing was performed using the Xpert HPV assay (Cepheid, Sunnyvale, CA, USA) which is a qualitative, real-time PCR assay for the detection of hrHPV DNA. The Xpert HPV assay includes simultaneous detection of 14 hrHPV types, hydroxymethylbilane synthase (HMBS) and an internal Probe Check Control. The 14 targeted hrHPV types are detected in five fluorescent channels: (i) HPV16; (ii) HPV18 and HPV45 (HPV18/45); (iii) HPV31, 33, 35, 52, and 58; (iv) HPV51 and HPV59; and (v) HPV39, 56, 66, and 68. HMBS (fluorescent channel 6) verifies specimen adequacy. Specimens were mixed and a 1-mL aliquot was removed using a disposable pipette and placed in the testing cartridge per the manufacturer's instructions. Unsatisfactory results due to insufficient cellular content were retested. If the second test was also unsatisfactory, the result was recorded as unsatisfactory [13].

## Analytic population

We used a team of two US gynecologic pathologists and one Rwandan general pathologist to review overlapping subsets of the 1,377 (97%) adequate biopsies out of 1,420 biopsies taken because of the limited availability of pathologists and recovery of the biopsies for external review: 1,373 (99.7%) in Rwanda, 1,370 (99.5%) at the Albert Einstein College of Medicine and 80 (5.8%) at Rutgers University for adjudication. We also conducted p16 immunohistochemistry (IHC) on 114 (8.3%) sections from biopsies to help clarify the diagnosis. We used the following conservative algorithm to assign a final diagnosis: 1) First, any diagnosis of unqualified high-grade squamous intraepithelial lesion (HSIL) or cervical intraepithelial neoplasia-CIN (HSIL unspecified or CIN2–3) was assigned a diagnosis of CIN2; 2) if there was only one pathology review and no p16 IHC data, that was the final diagnosis; 3) any CIN2 diagnosis by a gynecologic pathologist that was not confirmed at least by a second diagnosis of CIN2 by any pathologist or was p16 IHC negative was classified as <CIN2; 4) if there was more than one diagnosis, then we used the diagnosis(es) from the gynecologic pathologist(s) and used a diagnosis of CIN2+ by the general pathologist only as confirmation of a CIN2 diagnosis; 5) if there were diagnoses by both gynecologic pathologists, the worst diagnosis was assigned except if the worst diagnosis was CIN2 and there was no other confirmation of CIN2, then a < CIN2 diagnosis was assigned; and 6) if there was no biopsy taken or no adequate biopsy diagnosis but p16 IHC was adequate and called negative, a < CIN2 diagnosis was assigned. For this analysis, we excluded women with CIN2+ because the determinants of hrHPV positivity and women with CIN2 + could bias the sample due to collinearity of both outcomes and we did not want to conflate the determinants of HPV positivity with those of developing CIN2 + .

## Statistical analyses

Proportions were used to summarize the hypothesized risk factors. Continuous variables were categorized as follows: age (30–34, 35–39, 40–44, 45–49, and 50–54 years) and CD4 count as <200, 200–349, 350–499, and ≥500 cells/mL. We did not include viral load data in our analyses because they were only available for 28.4% of the participants and only for those with viral load measurements of ≤200 copies per milliliter according to national guidelines. We calculated the hrHPV prevalence and the binomial exact 95% confidence interval (95%CI). Distribution of hrHPV prevalence across categories of baseline characteristics (hypothesized risks factors) were compared using the Chi square, Fisher exact and Cochran-Armitage tests as appropriate. We stratified hrHPV prevalence by 1) age group, by CD4 count category, and both. We tested for trends across age groups within CD4 count category, and vice versa, using the Cochran-Armitage test.

We reported distribution of demographic, sexual and behavioral, pregnancy-related and HIV-related characteristics by hrHPV status (negative vs. positive) and then for hrHPV multi-level status (hrHPV negative vs. one-channel hrHPV positive vs. ≥ 2 hrHPV channels positive, the latter as a proxy for multiple HPV infections). Logistic regression models were used to estimate the crude odds ratios (OR) with 95%CI assessing the strength of the association between each risk factor and each outcome in the bivariate analyses. Multivariable logistic regression, including all variables found to be associated with the outcome in bivariate analyses, was used to calculate the adjusted odds ratios (aOR) with 95%CI assessing the strength of the independent association of the variable with the outcome. We re-did our model to see if some of these factors were more strongly associated with multi-channel hrHPV positivity, indicative of multiple hrHPV infections. For aORs, we also tested for linear trends across categories of variables. P values of <0.05 were considered statistically significant. STATA version 17.0 (STATA Corporation, College Station, TX, USA) was used for all analyses.

## Results

Of the 5,063 WLWH whose specimens were collected for HPV DNA testing, 4,955 (97.9%) had valid HPV results. Among the 1,631 screen-positive WLWH eligible for colposcopy, 1,420 (87.1%) underwent colposcopy. Among those, 1,377 (97%) had adequate biopsies (Fig 1). We excluded the 75 participants with CIN2+ (1.5% of 4,955), resulting in a final analysis group of 4,880.

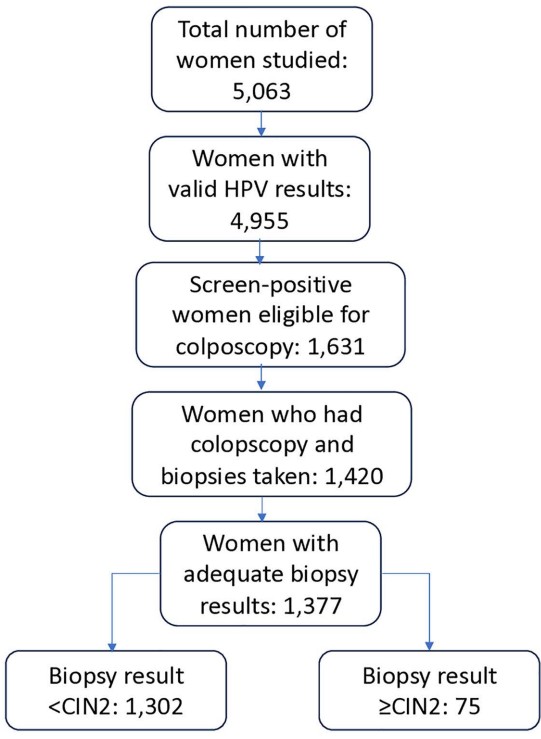

**Fig 1. Participant and lab results flow.**

Among those included, the mean (±standard deviation) age was 40.3 (±6.5) years. Over 98% WLWH were on antiretroviral therapy (ART) and 61% had a CD4 count of ≥500 (Table 1). Overall hrHPV prevalence was 25.5% (95% CI: 24.3%−26.8%); the positivity (prevalence) for HPV16, HPV18/45, HPV31/33/35/52/58, HPV 51/59 and HPV 39/56/66/68 was 5.4%, 5.4%, 13.5%, 3.5%, and 6.1%, respectively. Among hrHPV positives, 25.0% were positive for two or more HPV channels, 4.9% were positive for 3 or more channels, 1.4% were positive for 4 channels; and none were positive for all 5 channels.

There were strong trends for decreasing hrHPV prevalence with older age groups and higher CD4 count categories ($p_{trend}$<0.001 for both) (Fig 2) and likewise for continuous values of age and CD4 ($p_{trend}$<0.001 for both). There was decreasing hrHPV prevalence with older age groups for all CD4 count categories ($p_{trend}$≤0.001). There was also decreasing hrHPV prevalence for higher CD4 count categories for all age groups ($p_{trend}$<0.001) except the 50–54 years age group, for which there was no trend ($p_{trend}$ = 1.0).

Younger age, being single, separated or divorced, smaller household size, younger age at first sex, greater lifetime number of sexual partners, smoking, younger age at first pregnancy, ever using oral contraceptives, not being on ART, and lower CD4 cell counts were associated with hrHPV positivity (Table 1). Table 2 shows the results of a parsimonious, logistic regression model for hrHPV positivity for the 4,284 of 4,880 (87.8%) WLWH who had complete data for all included variables. There was no difference in hrHPV positivity between those included in the model compared to those not included in the model with 25.2% and 28.2%, respectively (p = 0.12). After mutual adjustment, younger age, smaller household size, increasing lifetime number of sexual partners, younger age at first pregnancy, ever using oral contraceptives and lower CD4 cell count were all significantly associated with hrHPV positivity, with $p_{trend}$ of <0.001 except for lifetime number of sexual partners ($p_{trend}$ 0.037). Being aged 30–34 years (vs. 50–54 years) (aOR=1.5; 95%CI = 1.1–1.9),

**Table 1. Baseline socio-demographic and clinical characteristics by hrHPV positivity (N = 4,880).**

| Characteristic type | Variables and categories | Overall: N = 4,880 n (col%) | hrHPV-: n = 3,634 (74.5%) n (col%) | hrHPV+: n = 1,246 (25.5%) n (col%) | p-value |
|---|---|---|---|---|---|
| Demographic | Age group (Years) | | | | **<0.001** |
| | 30-34 | 1,102 (22.6) | 749 (20.6) | 353 (28.3) | |
| | 35-39 | 1,295 (26.5) | 943 (26.0) | 352 (28.2) | |
| | 40-44 | 1,112 (22.8) | 867 (23.9) | 245 (19.7) | |
| | 45-49 | 868 (17.8) | 681 (18.7) | 187 (15.0) | |
| | 50-54 | 503 (10.3) | 398 (10.8) | 109 (8.8) | |
| | Marital status | | | | **<0.001** |
| | Married/Cohabiting | 2,463 (50.5) | 1,866 (51.4) | 597 (47.9) | |
| | Widowed | 1,008 (20.7) | 782 (21.5) | 226 (18.2) | |
| | Separated/Divorced | 791 (16.2) | 538 (14.8) | 253 (20.3) | |
| | Single | 538 (11.0) | 386 (10.6) | 152 (12.2) | |
| | Missing | 80 (1.6) | 62 (1.7) | 18 (1.4) | |
| | Province of origin | | | | 0.939 |
| | City of Kigali | 4,556 (93.4) | 3,396 (93.5) | 1,160 (93.1) | |
| | Eastern | 84 (1.7) | 63 (1.7) | 21 (1.7) | |
| | Northern | 35 (0.7) | 25 (0.7) | 10 (0.8) | |
| | Southern | 69 (1.4) | 48 (1.3) | 21 (1.7) | |
| | Western | 22 (0.5) | 17 (0.5) | 5 (0.4) | |
| | Missing | 114 (2.3) | 85 (2.3) | 29 (2.3) | |
| | Monthly income (FRW) | | | | 0.196 |
| | <40,000-40,000 | 2,508 (51.4) | 1,839 (50.6) | 669 (53.7) | |
| | 40,001-80,000 | 1,537 (31.5) | 1,167 (32.1) | 370 (29.7) | |
| | 80,001-120,000 | 382 (7.8) | 296 (8.2) | 86 (6.9) | |
| | ≥ 120,001 | 328 (6.7) | 243 (6.7) | 85 (6.8) | |
| | Missing | 125 (2.6) | 89 (2.4) | 36 (2.9) | |
| | Household size | | | | **0.001** |
| | 0-3 | 1,661 (34.1) | 1,190 (32.8) | 471 (37.8) | |
| | 4-6 | 2,368 (48.5) | 1,775 (48.8) | 593 (47.6) | |
| | ≥7 | 758 (15.5) | 601 (16.5) | 157 (12.6) | |
| | Missing | 93 (1.9) | 68 (1.9) | 25 (2.0) | |
| Sexual and behavioral | Age at first sex (Years) | | | | **<0.001** |
| | ≤ 16 | 1,089 (22.3) | 746 (20.5) | 343 (27.5) | |
| | 17-18 | 1,559 (32.0) | 1,141 (31.4) | 418 (33.6) | |
| | 19-20 | 973 (19.9) | 753 (20.7) | 220 (17.7) | |
| | ≥ 21 | 1,162 (23.8) | 920 (25.3) | 242 (19.4) | |
| | Missing | 97 (2.0) | 74 (2.1) | 23 (1.8) | |
| | Lifetime number of sexual partners | | | | **0.001** |
| | 0-1 | 768 (15.8) | 610 (16.8) | 158 (12.7) | |
| | 2-3 | 2,383 (48.8) | 1,784 (49.1) | 599 (48.1) | |
| | ≥ 4 | 1,402 (28.7) | 1,005 (27.6) | 397 (31.8) | |
| | Missing | 327 (6.7) | 235 (6.5) | 92 (7.4) | |
| | Number of sexual partners, last 6 months | | | | **<0.001** |
| | 0-1 | 4,273 (87.6) | 3,218 (88.6) | 1,055 (84.7) | |
| | 2-3 | 299 (6.1) | 205 (5.6) | 94 (7.5) | |
| | ≥ 4 | 146 (3.0) | 90 (2.5) | 56 (4.5) | |
| | Missing | 162 (3.3) | 121 (3.3) | 41 (3.3) | |
| | Smoking currently | | | | **0.020** |
| | No | 415 (8.5) | 315 (8.7) | 100 (8.0) | |
| | Yes | 82 (1.7) | 50 (1.4) | 32 (2.6) | |
| | Missing | 4,383 (89.8) | 3,269 (89.9) | 1,114 (89.4) | |

*(Continued)*

**Table 1.** (Continued)

| Characteristic type | Variables and categories | Overall: N=4,880 n (col%) | hrHPV-: n=3,634 (74.5%) n (col%) | hrHPV+: n=1,246 (25.5%) n (col%) | p-value |
|---|---|---|---|---|---|
| Pregnancy-related and contraception | Age at first pregnancy (Years) | | | | **<0.001** |
| | <18 | 1,397 (28.6) | 960 (26.4) | 437 (35.1) | |
| | 18-21 | 1,447 (29.7) | 1,077 (29.6) | 370 (29.7) | |
| | 22-24 | 905 (18.5) | 702 (19.3) | 203 (16.3) | |
| | ≥25 | 1,034 (21.2) | 820 (22.6) | 214 (17.2) | |
| | Missing | 97 (2.0) | 75 (2.1) | 22 (1.7) | |
| | Number of live births | | | | 0.183 |
| | 0-1 | 728 (14.9) | 524 (14.4) | 204 (16.4) | |
| | 2-4 | 2,931 (60.1) | 2,178 (59.9) | 753 (60.4) | |
| | ≥5 | 958 (19.6) | 728 (20.1) | 230 (18.5) | |
| | Missing | 263 (5.4) | 204 (5.6) | 59 (4.7) | |
| | Use of oral contraceptives (ever) | | | | **0.010** |
| | No | 3,175 (65.1) | 2,407 (66.2) | 768 (61.6) | |
| | Yes | 1,619 (33.2) | 1,162 (32.0) | 457 (36.7) | |
| | Missing | 86 (1.7) | 65 (1.8) | 21 (1.7) | |
| | Condom use (ever) | | | | 0.384 |
| | No | 2,937 (60.2) | 2,172 (59.8) | 765 (61.4) | |
| | Yes | 1,850 (37.9) | 1,388 (38.2) | 462 (37.1) | |
| | Missing | 93 (1.9) | 74 (2.0) | 19 (1.5) | |
| HIV-related data (Medical records) | On ART | | | | **0.081** |
| | Yes | 4,744 (97.2) | 3,541 (97.5) | 1,203 (96.6) | |
| | No | 60 (1.2) | 37 (1.0) | 23 (1.8) | |
| | Missing | 76 (1.6) | 56 (1.5) | 20 (1.6) | |
| | CD4 cell count | | | | **<0.001** |
| | <200 | 256 (5.2) | 143 (3.9) | 113 (9.1) | |
| | 200-349 | 529 (10.8) | 354 (9.8) | 175 (14.0) | |
| | 350-499 | 1,024 (21.0) | 775 (21.3) | 249 (20.0) | |
| | ≥500 | 2,823 (57.9) | 2,184 (60.1) | 639 (51.3) | |
| | Missing | 248 (5.1) | 178 (4.9) | 70 (5.6) | |

having a CD4 count of <200 per mL (vs. ≥500 per mL) (aOR=2.7, 95%CI=2.1–3.6), and age at first pregnancy younger than 18 years (vs. ≥25 years) (aOR=1.7; 95%CI=1.4–2.1) were the individual factors most strongly associated with being hrHPV positive.

Distributions of risk factors by hrHPV multi-level status [negative (74.2%), single-hrHPV channel positive (22.3%), and multi-channel-hrHPV positive (3.2%)] shown in Table 3 were similar to those shown in Table 1 for hrHPV positivity, except that number of live births and monthly income were weakly associated with hrHPV multi-level positivity. After mutual adjustment in a logistic regression model, the factors associated with hrHPV positivity remained associated with single- and multi-channel-hrHPV positivity although the associations were qualitatively stronger for age, age at first pregnancy and CD4 counts. Notably, having a CD4 count of <200 per mL was very strongly associated with being multi-channel hrHPV positivity (aOR=4.8; 95%CI=2.8–8.2), Table 4.

## Discussion

Here, we report on the prevalence of hrHPV and the association between various socio-demographic and clinical characteristics with testing hrHPV positive overall and for multiple hrHPV channels as a measure for multiple hrHPV infections.

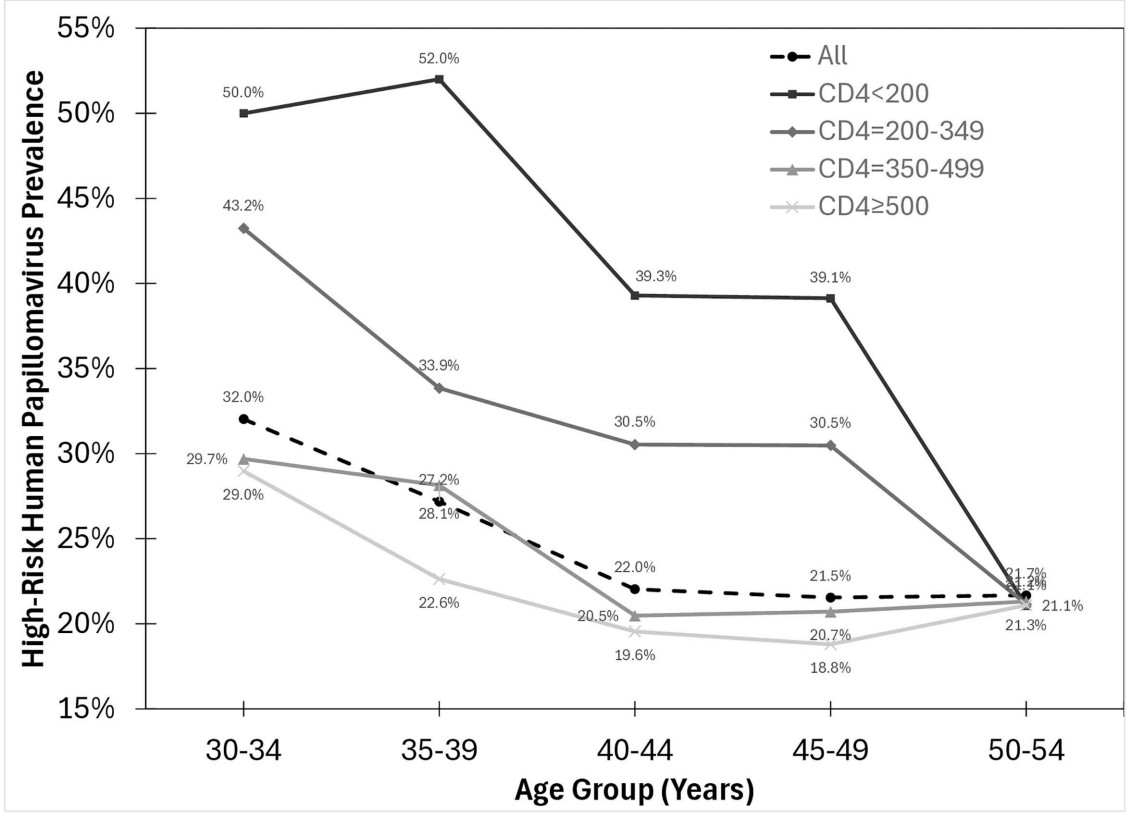

**Fig 2. High-risk human papillomavirus prevalence stratified by age group and CD4 count category.** The high-risk human papillomavirus prevalence was 25.4% overall and was 44.14%, 33.08%, 24.32%, and 22.64% for those with a CD4 count <200, 200-349, 350-499, and 500+ per mL, respectively.

The strongest factor for testing hrHPV positive was the concurrent CD4 count, with those having the lowest counts and the most immune suppressed having the highest prevalence of hrHPV (44.1% for CD4 counts of <200 per mL vs. 22.6% for CD4 counts of ≥500 per mL) and the most likely to test positive for multiple hrHPV channels (7.8% for CD4 counts of <200 per mL vs. 2.5% for CD4 counts of ≥500 per mL). These findings are consistent with our previous report on the trend of decreasing hrHPV prevalence among Rwandan WLWH over time as more WLWH accessed ART and their immune status improved [15]; at the time of this study, only 5.5% of women had a CD4 count of <200 per mL. This highlights the role of HIV care in hrHPV prevalence and possibly cervical cancer risk.

However, WLWH remain at increased risk for cervical cancer [16,17] and screening them with a high-performance test, HPV DNA testing, will continue to be important if we are to achieve the WHO targets for the accelerated elimination of cervical cancer as a public health problem [5]. Our finding of 25.5% hrHPV prevalence is lower than that found in other studies [18–22] which found hrHPV prevalence rates as high as over 50%. This is most likely due to the high ART coverage in Rwanda and the successful HIV care program as indicated by the decrease in hrHPV prevalence over time among Rwandan WLWH [15]. However, a study of 221 WLWH living in Togo reported a lower prevalence of hrHPV than that found in our study [23].

We found several sociodemographic and clinical characteristics to be associated with hrHPV positivity. These findings are like those reported in other studies [18,20,22]. Nonetheless, there is variability in findings for factors associated with hrHPV with some studies finding associations with some of the factors such as a high number of sex partners and

**Table 2. Logistic regression model (N = 4,284) for variables associated with hrHPV positivity.**

| Variable | OR | 95% CI | aOR | 95% CI |
|---|---|---|---|---|
| Age group (Years) | | | | |
| 30-34 | 1.7 | 1.3-2.2 | **1.5** | 1.1-1.9 |
| 35-39 | 1.4 | 1.1-1.7 | 1.2 | 0.9-1.6 |
| 40-44 | 1.0 | 0.8-1.3 | 0.9 | 0.7-1.2 |
| 45-49 | 1.0 | 0.8-1.3 | 1.0 | 0.7-1.3 |
| 50-54 | Ref. | – | – | – |
| P$_{trend}$ | | | <0.001 | |
| Household size | | | | |
| 0-3 | 1.3 | 1.2-1.9 | **1.5** | 1.2-1.8 |
| 4-6 | 1.5 | 1.0-1.6 | **1.3** | 1.0-1.6 |
| ≥7 | Ref. | – | – | – |
| P$_{trend}$ | | | <0.001 | |
| Lifetime number of sexual partners | | | | |
| 0-1 | Ref. | – | – | – |
| 2-3 | 1.3 | 1.1-1.6 | 1.2 | 0.9-1.5 |
| ≥4 | 1.5 | 1.2-1.9 | **1.3** | 1.0-1.6 |
| P$_{trend}$ | | | 0.037 | |
| Age at first pregnancy (Years) | | | | |
| <18 | 1.7 | 1.4-2.1 | **1.7** | 1.4-2.1 |
| 18-21 | 1.3 | 1.1-1.6 | **1.3** | 1.1-1.6 |
| 22-24 | 1.1 | 0.9-1.4 | 1.1 | 0.9-1.4 |
| ≥25 | Ref. | – | – | – |
| P$_{trend}$ | | | <0.001 | |
| Use of oral contraceptives (ever) | | | | |
| No | Ref. | – | – | – |
| Yes | 1.2 | 1.1-1.4 | **1.3** | 1.1-1.5 |
| CD4 cell count | | | | |
| <200 | 2.7 | 2.1-3.5 | **2.7** | 2.1-3.6 |
| 200-349 | 1.7 | 1.4-2.1 | **1.7** | 1.4-2.2 |
| 350-499 | 1.1 | 0.9-1.3 | **1.2** | 1.0-1.4 |
| ≥500 | Ref. | – | – | – |
| P$_{trend}$ | | | <0.001 | |

lower CD4 counts [20] and others not finding any significant associations after comparing hrHPV with age, marital status, socioeconomic status, level of education, number of pregnancies, HIV viral load and CD4 cell count [23]. Some of these associations were observed to be stronger for multi-channel hrHPV positivity that may be indicative of/a proxy for riskier sexual behaviors leading to hrHPV acquisition (e.g., younger age) or viral persistence (e.g., lower CD4 counts).

Surprisingly, the decreasing hrHPV positivity with increasing CD4 cell counts was not seen in the oldest age group (50–54 years). We do not have an explanation of this observation, which might just be the result of small numbers in this group. An alternative possibility is a survival bias in which those older women with lower CD4 counts lived longer with HIV because of a healthier immune system than others who did not and they were better able to clear/control hrHPV.

Our study has limitations that deserve mention. First, the assay we used for HPV DNA testing in which HPV types, except for HPV16, were detected as pools of types. We could not study HPV type-specific patterns or know the exact number of HPV types present. Second, we had a lower-than-expected prevalence of CIN2+ in our population, as well as some loss-to-follow-up of women who may have had CIN2+, leading to some misclassification of women and conflating risk factors for hrHPV positivity and having CIN2+ among those who were hrHPV positive. However, these numbers are relatively small and likely only impacted the magnitude of the estimates, not the direction of the association. Additionally,

**Table 3. Baseline socio-demographic and clinical characteristics by multi-level hrHPV (N = 4,880).**

| Characteristic type | Variables and categories | Negative: n = 3,634 (74.5%) n (col%) | One channel+: n = 1,089 (22.3%) n (col%) | ≥2 channels+: n = 157 (3.2%) n (col%) | p-value |
|---|---|---|---|---|---|
| Demographic | Age group (Years) | | | | **<0.001** |
| | 30-34 | 749 (20.6) | 298 (27.4) | 55 (35.0) | |
| | 35-39 | 943 (26.0) | 308 (28.3) | 44 (28.0) | |
| | 40-44 | 867 (23.9) | 216 (19.8) | 29 (18.5) | |
| | 45-49 | 681 (18.7) | 168 (15.4) | 19 (12.1) | |
| | 50-54 | 394 (10.8) | 99 (9.1) | 10 (6.4) | |
| | Marital status | | | | **<0.001** |
| | Married/Cohabiting | 1,866 (51.4) | 539 (49.5) | 58 (36.9) | |
| | Widowed | 782 (21.5) | 192 (17.6) | 34 (21.7) | |
| | Separated/Divorced | 538 (14.8) | 216 (19.8) | 37 (23.6) | |
| | Single | 386 (10.6) | 126 (11.6) | 26 (16.5) | |
| | Missing | 62 (1.7) | 16 (1.5) | 2 (1.3) | |
| | Province of origin | | | | 0.995 |
| | City of Kigali | 3,396 (93.5) | 1,013 (93.0) | 147 (93.6) | |
| | Eastern | 63 (1.7) | 19 (1.7) | 2 (1.3) | |
| | Northern | 25 (0.7) | 9 (0.8) | 1 (0.6) | |
| | Southern | 48 (1.3) | 18 (1.7) | 3 (1.9) | |
| | Western | 17 (0.5) | 5 (0.5) | 0 (0.0) | |
| | Missing | 85 (2.3) | 25 (2.3) | 4 (2.6) | |
| | Monthly income (FRW) | | | | **0.063** |
| | <40,000-40,000 | 1,839 (50.6) | 569 (52.3) | 100 (63.7) | |
| | 40,001-80,000 | 1,167 (32.1) | 337 (30.9) | 33 (21.0) | |
| | 80,001-120,000 | 296 (8.2) | 78 (7.2) | 8 (5.1) | |
| | ≥120,001 | 243 (6.7) | 73 (6.7) | 12 (7.6) | |
| | Missing | 89 (2.4) | 32 (2.9) | 4 (2.6) | |
| | Household size | | | | **0.002** |
| | 0-3 | 1,190 (32.8) | 401 (36.8) | 70 (44.6) | |
| | 4-6 | 1,775 (48.8) | 530 (48.7) | 63 (40.1) | |
| | ≥7 | 601 (16.5) | 136 (12.5) | 21 (13.4) | |
| | Missing | 68 (1.9) | 22 (2.0) | 3 (1.9) | |
| Sexual and behavioral | Age at first sex (Years) | | | | **<0.001** |
| | ≤16 | 746 (20.5) | 279 (27.3) | 46 (29.3) | |
| | 17-18 | 1,141 (31.4) | 358 (32.9) | 60 (38.2) | |
| | 19-20 | 753 (20.7) | 195 (17.9) | 25 (15.9) | |
| | ≥21 | 920 (25.3) | 219 (20.1) | 23 (14.7) | |
| | Missing | 74 (2.1) | 20 (1.8) | 3 (1.9) | |
| | Lifetime number of sexual partners | | | | **0.003** |
| | 0-1 | 610 (16.8) | 139 (12.8) | 19 (12.1) | |
| | 2-3 | 1,784 (49.1) | 532 (48.9) | 67 (42.7) | |
| | ≥4 | 1,005 (27.6) | 339 (31.1) | 58 (36.9) | |
| | Missing | 235 (6.5) | 79 (7.2) | 13 (8.3) | |
| | Number of sexual partners, last 6 months | | | | **<0.001** |
| | 0-1 | 3,218 (88.6) | 932 (85.6) | 123 (78.4) | |
| | 2-3 | 205 (5.6) | 80 (7.4) | 14 (8.9) | |
| | ≥4 | 90 (2.5) | 44 (4.0) | 12 (7.6) | |
| | Missing | 121 (3.3) | 33 (3.0) | 8 (5.1) | |
| | Smoking currently | | | | **0.067** |
| | No | 315 (8.7) | 87 (8.0) | 13 (8.3) | |
| | Yes | 50 (1.4) | 27 (2.5) | 5 (3.2) | |
| | Missing | 3,269 (89.9) | 975 (89.5) | 139 (88.5) | |

*(Continued)*

**Table 3.** (Continued)

| Characteristic type | Variables and categories | Negative: n=3,634 (74.5%) n (col%) | One channel+: n=1,089 (22.3%) n (col%) | ≥2 channels+: n=157 (3.2%) n (col%) | p-value |
|---|---|---|---|---|---|
| Pregnancy-related and contraception | Age at first pregnancy (Years) | | | | **<0.001** |
| | < 18 | 960 (26.4) | 377 (34.6) | 60 (38.2) | |
| | 18-21 | 1,077 (29.6) | 318 (29.2) | 52 (33.1) | |
| | 22-24 | 702 (19.3) | 181 (16.6) | 22 (14.0) | |
| | ≥ 25 | 820 (22.6) | 194 (17.8) | 20 (12.8) | |
| | Missing | 75 (2.1) | 19 (1.8) | 3 (1.9) | |
| | Number of live births | | | | 0.021 |
| | 0-1 | 524 (14.4) | 166 (15.2) | 38 (24.2) | |
| | 2-4 | 2,178 (60.0) | 672 (61.7) | 81 (51.6) | |
| | ≥ 5 | 728 (20.0) | 201 (18.5) | 29 (18.5) | |
| | Missing | 204 (5.6) | 50 (4.6) | 9 (5.7) | |
| | Use of oral contraceptives (ever) | | | | **0.044** |
| | No | 2,407 (66.2) | 669 (61.4) | 99 (63.1) | |
| | Yes | 1,162 (32.0) | 402 (36.9) | 55 (35.0) | |
| | Missing | 65 (1.8) | 18 (1.7) | 3 (1.9) | |
| | Condom use (ever) | | | | 0.725 |
| | No | 2,172 (59.8) | 665 (61.1) | 100 (63.7) | |
| | Yes | 1,388 (38.2) | 407 (37.4) | 55 (35.0) | |
| | Missing | 74 (2.0) | 17 (1.5) | 2 (1.3) | |
| HIV-related data (Medical records) | On ART | | | | **0.035** |
| | Yes | 3,541 (97.5) | 1,053 (96.7) | 150 (95.6) | |
| | No | 37 (1.0) | 17 (1.6) | 6 (3.8) | |
| | Missing | 56 (1.5) | 19 (1.7) | 1 (0.6) | |
| | CD4 cell count | | | | **<0.001** |
| | < 200 | 143 (3.9) | 93 (8.5) | 20 (12.8) | |
| | 200-349 | 354 (9.8) | 147 (13.5) | 28 (17.8) | |
| | 350-499 | 775 (21.3) | 221 (20.3) | 28 (17.8) | |
| | ≥ 500 | 2,184 (60.1) | 568 (52.2) | 71 (45.2) | |
| | Missing | 178 (4.9) | 60 (5.5) | 10 (6.4) | |

we were not able to include HIV viral load data because most women did not have available viral load results. However, data on CD4 cell counts were sufficiently available and they could serve as a surrogate for HIV disease status.

Our study has strengths as well. This is among the few large studies reporting on the determinants of hrHPV positivity in WLWH conducted anywhere and especially in Rwanda and SSA. We collected data on factors known to be associated with hrHPV positivity and cervical cancer that made this study comprehensive. Although our disease ascertainment was likely imperfect, we did identify and were able to exclude many women with CIN2+ whereas some other studies conducted in SSA relied on other, insensitive measures of disease such as visual inspection or cytologic interpretation [19,24–27].

## Conclusions

Our findings highlight the role of CD4 counts, as a measure of HIV control and immunity, on controlling hrHPV infection, which could potentially impact cervical cancer risk. Multiple factors might also contribute to hrHPV infection and persistence. Raising awareness on those factors coupled with integrating HPV and cervical cancer awareness in HIV care could help control this double burden of disease.

**Table 4. Logistic regression model (N=4,284) for variables associated with multi-level hrHPV positivity (Base outcome: Negative).**

| Variable: | One channel positive | | ≥2 channels positive | |
|---|---|---|---|---|
| | OR (95% CI) | aOR (95% CI) | OR (95% CI) | aOR (95% CI) |
| Age group (Years) | | | | |
| 30-34 | 1.6 (1.2-2.0) | **1.3 (1.0-1.8)** | 2.9 (1.5-5.7) | **2.3 (1.1-4.8)** |
| 35-39 | 1.3 (1.0-1.7) | 1.2 (0.9-1.6) | 1.8 (0.9-3.7) | 1.5 (0.7-3.3) |
| 40-44 | 1.0 (0.8-1.3) | 0.9 (0.7-1.2) | 1.3 (0.6-2.7) | 1.2 (0.5-2.6) |
| 45-49 | 1.0 (0.7-1.3) | 1.0 (0.7-1.3) | 1.1 (0.5-2.4) | 1.1 (0.5-2.6) |
| 50-54 | Ref. | Ref. | Ref. | Ref. |
| $P_{trend}$ | | | <0.001 | |
| Household size | | | | |
| 0-3 | 1.5 (1.2-1.9) | **1.4 (1.1-1.8)** | 1.7 (1.0-2.8) | **1.7 (1.0-3.0)** |
| 4-6 | 1.3 (1.1-1.6) | **1.3 (1.0-1.6)** | 1.3 (0.6-1.7) | 1.0 (0.6-1.8) |
| ≥7 | Ref. | Ref. | Ref. | Ref. |
| $P_{trend}$ | | | <0.001 | |
| Lifetime number of sexual partners | | | | |
| 0-1 | Ref. | Ref. | Ref. | Ref. |
| 2-3 | 1.3 (1.1-1.6) | **1.2 (1.0-1.5)** | 1.2 (0.7-2.0) | 1.0 (0.6-1.8) |
| ≥4 | 1.5 (1.2-1.8) | **1.3 (1.0-1.6)** | 1.8 (1.1-3.1) | 1.3 (0.7-2.3) |
| $P_{trend}$ | | | 0.0001 | |
| Age at first pregnancy (Years) | | | | |
| <18 | 1.7 (1.4-2.0) | **1.6 (1.3-2.0)** | 2.6 (1.5-4.3) | **3.0 (1.6-5.4)** |
| 18-21 | 1.2 (1.0-1.5) | **1.2 (1.0-1.5)** | 2.0 (1.2-3.3) | **2.5 (1.3-4.5)** |
| 22-24 | 1.1 (0.9-1.4) | 1.1 (0.8-1.4) | 1.3 (0.7-2.4) | 1.8 (0.9-3.5) |
| ≥25 | Ref. | Ref. | Ref. | Ref. |
| $P_{trend}$ | | | <0.001 | |
| Use of oral contraceptives (ever) | | | | |
| No | Ref. | Ref. | Ref. | Ref. |
| Yes | 1.2 (1.1-1.4) | **1.2 (1.1-1.4)** | 1.2 (0.8-1.6) | 1.3 (0.9-1.9) |
| CD4 cell count | | | | |
| <200 | 2.5 (1.9-3.3) | **2.5 (1.9-3.4)** | 4.3 (2.5-7.3) | **4.8 (2.8-8.2)** |
| 200-349 | 1.6 (1.3-2.0) | **1.7 (1.3-2.1)** | 2.4 (1.5-3.8) | **2.6 (1.6-4.2)** |
| 350-499 | 1.1 (0.9-1.3) | 1.2 (0.97-1.4) | 1.1 (0.7-1.7) | 1.3 (0.8-2.0) |
| ≥500 | Ref. | Ref. | Ref. | Ref. |
| $P_{trend}$ | | | <0.001 | |

## Supporting information

**S1 File. Inclusivity-in-global-research-questionnaire_filed_Murenzi.**
(PDF)

## Acknowledgments

We acknowledge the study participants for their contribution to the study as well as the study staff including nurses, laboratory technicians and administrative and finance staff. We are also grateful for the collaborating sites or health centers at which we did cervical cancer screening for their support towards the success of the study.

## Author contributions

**Conceptualization:** Kathryn Anastos, Philip E. Castle.

**Data curation:** Maria Demarco, Benjamin Muhoza, Kristen Hansen, Julia C. Gage.

**Formal analysis:** Gad Murenzi, Philip E. Castle.

**Funding acquisition:** Adebola Adedimeji, Marcel Yotebieng, Leon Mutesa, Kathryn Anastos.

**Investigation:** Gad Murenzi, Faustin Kanyabwisha, Jean Paul Mivumbi, Anthere Murangwa, Theogene Rurangwa, Thierry Muvunyi Zawadi, Tiffany Hébert.

**Methodology:** Marcel Yotebieng, Philip E. Castle.

**Project administration:** Gallican Kubwimana, Kathryn Anastos.

**Software:** Maria Demarco, Benjamin Muhoza, Kristen Hansen, Philip E. Castle.

**Supervision:** Gad Murenzi, Theogene Rurangwa, Gallican Kubwimana, Julia C. Gage, Laetitia Nyirazinyoye, Leon Mutesa, Kathryn Anastos, Philip E. Castle.

**Validation:** Faustin Kanyabwisha, Tiffany Hébert.

**Visualization:** Gad Murenzi.

**Writing – original draft:** Gad Murenzi, Philip E. Castle.

**Writing – review & editing:** Faustin Kanyabwisha, Maria Demarco, Benjamin Muhoza, Kristen Hansen, Jean Paul Mivumbi, Anthere Murangwa, Theogene Rurangwa, Thierry Muvunyi Zawadi, Gallican Kubwimana, Julia C. Gage, Tiffany Hébert, Adebola Adedimeji, Laetitia Nyirazinyoye, Marcel Yotebieng, Leon Mutesa, Kathryn Anastos, Philip E. Castle.

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
