## [Decision Letter · Decision Letter 0]

9 Jul 2025

PONE-D-25-13630Determinants of cervical high-risk human papillomavirus positivity among Rwandan women with human immunodeficiency virusPLOS ONE

Dear Dr. Murenzi,

Thank you for submitting your manuscript to PLOS ONE. After careful consideration, we feel that it has merit but does not fully meet PLOS ONE’s publication criteria as it currently stands. Therefore, we invite you to submit a revised version of the manuscript that addresses the points raised during the review process. The reviewers have offered editorial suggestions, which you may consider, and questions aboout methods and rationale for analysis which should be answered. Please look carefully at the reviewers’ comments, and revise or respond as appropriate. The succinctness of this paper is admirable. However, it would be helpful if you could provide more rationale with regard to a few issues.

Please ensure that the following issues are clearly explained:

The selection of sample size.

The decision to exclude women with CIN2+ from the study.

The decision to merge the data for 2 channels and >2 channels positive in analysing ‘multilevel positivity’.

Is it possible to comment or provide data on the acceptance rate for the study among women who were eligible for or were offered HPV testing?

Reviewer 2 has asked a number of questions about study design. Please review these carefully and, if it is not possible to answer the question directly, consider discussing these issues as study limitations.One question, regarding use of the multilevel data, may have already been covered in your manuscript. If so, please ensure that the contribution of the multilevel data to the analyses is clearly described, and that the results are highlighted in the discussion.

One issue raised by both reviewers is that of viral load data and its potential use as a predictor or correlate. Please comment on whether such data would be useful, given the high rate of compliance with ARV treatment. If so, pllease consider including the data.  In some situations in Africa, viral load is used more than CD4 for immune monitoring; will CD4 data be available going forward in Rwanda, and if not, what is the implication.

We look forward to receiving your revised manuscript.

Kind regards,

Patricia Evelyn Fast, MD, Ph.D.

Academic Editor

PLOS ONE

Journal Requirements:

5. In the online submission form, you indicated that data cannot be shared publicly because of restrictions about data sharing by the Rwanda National Ethics Committee and the Government of Rwanda.

6. We note that the grant information you provided in the ‘Funding Information’ and ‘Financial Disclosure’ sections do not match.

7. We note you have included a table to which you do not refer in the text of your manuscript. Please ensure that you refer to Table 4 in your text; if accepted, production will need this reference to link the reader to the Table.

Reviewers' comments:

Reviewer's Responses to Questions

**Comments to the Author**

1. Is the manuscript technically sound, and do the data support the conclusions?

Reviewer #1: Yes

Reviewer #2: Partly

2. Has the statistical analysis been performed appropriately and rigorously? 

Reviewer #1: Yes

Reviewer #2: Yes

3. Have the authors made all data underlying the findings in their manuscript fully available?

Reviewer #1: Yes

Reviewer #2: Yes

4. Is the manuscript presented in an intelligible fashion and written in standard English?

Reviewer #1: Yes

Reviewer #2: Yes

5. Review Comments to the Author

Reviewer #1: This is a well written paper.

Title

The title of the manuscript should read: Determinants of cervical high-risk human papillomavirus positivity among Rwandan women living with human immunodeficiency virus.

Abstract

Here, authors stated that they screened women from 2016 to 2020, but in the paper, but line 102 shows they screened participants from 2016 to 2018. Authors should please state the reasons for the discrepancy.

Ethics Statement

Authors should please include the ethical clearance reference number.

Material and Methods

Since the study is aimed at identifying key determinants of hrHPV among women living with HIV, it is unclear why the authors did not include women uninfected with HIV as control.

Authors mentioned that the collected data on viral load (line 117) but presented none. Was it an omission?

How was the CD4 T cell counted? Authors should state the controls used for the CD4 T cell count and HPV DNA testing.

Results

What was the overall prevalence of hrHPV among women diagnosed of <cin and="" cin2="">

Discussion

Authors alluded that the lower prevalence of hrHPV in Rwanda is because of ART coverage in Rwanda. Did the authors assess the rate of participants’ compliance to ART regime in their study and the studies they referenced? It is unclear if the authors assess HPV vaccination among the participants as it could impact on the prevalence rate of hrHPV.</cin>

Reviewer #2: 1. This study addresses a crucial public health issue: the intersection between HIV infection and cervical cancer risk, particularly through hrHPV infection. The authors tackle this issue with impressive breadth, drawing attention to a vulnerable population in a region where data is urgently needed. But as I read the introduction, I wonder — could the research question be more explicitly stated? While the rationale is strong, the specific hypothesis or aim feels somewhat implied rather than clearly declared.

2. The connection drawn between the WHO’s 90-70-90 targets and the reality of hrHPV testing availability in sub-Saharan Africa is compelling. The authors rightfully highlight the gap between global goals and local capacities. However, I am left wondering: how was Rwanda’s specific readiness for HPV DNA testing evaluated before the 2019 screening program launch? More context on this local implementation process would strengthen the narrative.

3. The methodology stands out for its rigor. The combination of HPV DNA testing, VIA, and confirmatory colposcopy with biopsy is robust and aligns with best practices. I appreciate the detail provided on lab testing and the conservative diagnostic algorithm. Still, does the use of pooled hrHPV typing (except for HPV16) limit our ability to identify the specific subtypes driving disease in this population?

4. The sample size — over 5,000 women — is remarkable and lends strong credibility to the findings. The demographic spread across all Rwandan provinces increases generalizability. Yet, I’m curious: were there any regional differences in hrHPV prevalence or immune status that could be relevant for tailoring interventions?

5. The data collection methods are thorough, combining nurse-administered questionnaires with EMR extraction. This mixed approach likely improves completeness, but could the reliance on self-reported sexual and reproductive history introduce recall bias? How might this have influenced the associations observed?

6. The statistical analyses are appropriately complex and well-executed. Stratified analyses by age and CD4 count bring valuable nuance. That said, the decision to exclude women with CIN2+ from the final analysis raises questions. Could this exclusion have masked relevant associations between risk factors and disease progression?

7. One of the most striking findings is the strong inverse association between CD4 count and hrHPV positivity, particularly for multiple HPV channel infections. This points to immunosuppression as a major driver of viral persistence. But it also prompts a question: could ART duration or viral suppression levels provide further insight into this relationship?

8. The multivariable model is well-constructed and interpretable, with adjusted odds ratios that clearly illustrate the strength of associations. However, I wonder whether any interaction terms (e.g., between age and CD4 count, or ART use and number of partners) were explored? Such analyses could reveal more complex dynamics at play.

9. The authors report an overall hrHPV prevalence of 25.5%, which is lower than in many other SSA studies. This is attributed to Rwanda’s successful ART program. It’s a hopeful sign, but could selection bias (due to missing or excluded data) partially explain the lower prevalence?

10. The paper observes that in women aged 50–54, the expected decrease in hrHPV prevalence with higher CD4 counts was not observed. This is acknowledged but left somewhat unexplored. Could this be due to survival bias? Or might other factors like cervical epithelial changes with age play a role?

11. The findings regarding sexual behavior — such as number of partners and age at first sex — being associated with hrHPV positivity are not surprising, but still significant. I do wonder, though: to what extent are these factors modifiable in this population? How can these findings inform effective, culturally sensitive prevention strategies?

12. Oral contraceptive use was also associated with increased hrHPV positivity. While biologically plausible, this association warrants further exploration. Could this be a proxy for unprotected sex or other behaviors not fully captured in the dataset?

13. The authors mention that monthly income and number of live births were only weakly associated with multi-channel hrHPV positivity. Yet socioeconomic status often plays a critical role in health outcomes. Could this apparent lack of association reflect limitations in how income was measured?

14. I appreciated the transparency in discussing the study’s limitations. The lack of type-specific HPV analysis due to pooled testing is a clear constraint. Might future studies in Rwanda benefit from using assays that provide more granular typing?

15. Another limitation — the relatively low CIN2+ prevalence and loss to follow-up — could influence both the interpretation and generalizability of findings. Could a follow-up study or cohort design provide better insight into progression risk and persistence?

16. The ethical conduct of the study is commendable, especially considering the sensitive nature of both HIV and HPV status. Participant protection and data integrity appear to have been taken seriously throughout.

17. The discussion of how findings relate to WHO elimination goals is strong. Still, I was hoping for more concrete policy recommendations. For instance, should screening intervals for WWH be adjusted based on CD4 count? What would an ideal integrated HIV-HPV care model look like in Rwanda?

18. A humanizing aspect of this study is how it captures the lived intersection of HIV and women’s health. Beyond statistical significance, the findings have real-world implications: each data point represents a woman at risk, navigating systems that may not always prioritize her dual burden of disease.

19. While this paper is methodologically sound and clinically relevant, it opens up several unanswered questions. Could longitudinal follow-up reveal more about which WWH are most at risk for progression to CIN2+ or cancer? What role does HPV vaccination have in this context — especially among younger HIV-positive women?

20. In conclusion, this study is a valuable contribution to cervical cancer research in HIV-positive populations. It brings both solid evidence and new questions to the table. The challenge now is to translate these findings into action — improving screening, investing in accessible HPV testing, and integrating prevention strategies into HIV care, all while continuing to listen to and understand the women at the center of these data.

6. PLOS authors have the option to publish the peer review history of their article (what does this mean? ). If published, this will include your full peer review and any attached files.

**Do you want your identity to be public for this peer review?** For information about this choice, including consent withdrawal, please see our Privacy Policy .

Reviewer #1: **Yes: ** Jude Ogechukwu Okoye

Reviewer #2: **Yes: ** Jonas Michel Wolf

---

## [Author Response · Author response to Decision Letter 1]

25 Aug 2025

Gad Murenzi, MD, MPH, MMed

Einstein-Rwanda Research and Capacity Building Program

Research for Development (RD Rwanda)

Kigali, Rwanda

Email: gadcollins@gmail.com

The Editors, PLOS One

Suite 103 #188, 1875 Mission Street

San Francisco, CA 94103

United States

August 23, 2025

RE: Response to Reviewers’ comments

Thank you so much for accepting to review our manuscript titled; “Determinants of cervical high-risk human papillomavirus positivity among Rwandan women with human immunodeficiency virus”. We think that the reviews and feedback will make our manuscript stronger. Below is a point-by-point response to the reviewers’ comments and questions;

Academic editor’s comments

Please ensure that the following issues are clearly explained:

The selection of sample size.

Response: A sentence has been added to clarify the sample size selection (lines 106-107 on pages 5-6) but given that the study protocol has been published (https://pubmed.ncbi.nlm.nih.gov/30082342/), we refer to it for further details about sample size calculations.

The decision to exclude women with CIN2+ from the study.

Response: We excluded women with CIN2+ because of collinearity between that outcome and major outcome of this analysis (hrHPV positivity) and a sentence has been added to clarify it (lines 160-163 on page 8). Also, we excluded CIN2+ from the analysis to make our analysis comparable to the other main studies of HPV prevalence and determinants, such as those published from IARC and the NCI. Traditionally, the CIN2+ has been excluded to avoid conflating determinants of HPV infection with those of progression to CIN2+. However, our estimates are unlikely to change significantly as most HPV positives (~90%) do not have concurrent CIN2+.

The decision to merge the data for 2 channels and >2 channels positive in analysing ‘multilevel positivity’.

Response: There are relatively few women who are positive for 3 or more channels so it made sense to us to combine all multiple channel positivity into one category, creating three tiers: hrHPV negative, hrHPV positive for one channel, and hrHPV positive for two or more channels. We have added the proportions in the text to indicate the low positivity for ≥2 channels (lines 232-233 on page 11).

Is it possible to comment or provide data on the acceptance rate for the study among women who were eligible for or were offered HPV testing?

Response: We enrolled women from 15 HIV clinics and the number of WLWH aged 30-54 years at each clinic varied. We tried to enroll all eligible women at each clinic and although we did not objectively keep record of how many women declined screening, our experience indicates that most eligible women who were offered screening accepted it and volunteered to participate in the study. Additionally, this study was conducted at a time when HPV -based screening was not available in Rwanda hence the high participation.

Reviewer 2 has asked a number of questions about study design. Please review these carefully and, if it is not possible to answer the question directly, consider discussing these issues as study limitations.One question, regarding use of the multilevel data, may have already been covered in your manuscript. If so, please ensure that the contribution of the multilevel data to the analyses is clearly described, and that the results are highlighted in the discussion.

Response: Thank you for the comment, we have provided relevant details and responses to the reviewers’ comments.

One issue raised by both reviewers is that of viral load data and its potential use as a predictor or correlate. Please comment on whether such data would be useful, given the high rate of compliance with ARV treatment. If so, pllease consider including the data. In some situations in Africa, viral load is used more than CD4 for immune monitoring; will CD4 data be available going forward in Rwanda, and if not, what is the implication.

Response: We agree that in most African settings, CD4 count is no longer widely available and viral load (VL) testing is more available. However, CD4 count continues to be a valuable measure for advanced HIV disease especially at HIV diagnosis with potential to inform HPV and cervical cancer prevention and control. We also know that higher CD4 cell counts play a crucial role in HPV control in terms of prevalence and persistence hence potential for cervical cancer control. The other issue is that very few women had available VL data hence limited power to make comparisons. A sentence about this has been added to the Discussion section under limitations (lines 291-293 on page 13).

Reviewers' Comments

Reviewer #1

This is a well written paper.

Response: Thank you and we appreciate the reviewer for taking their time to review our paper.

Title

The title of the manuscript should read: Determinants of cervical high-risk human papillomavirus positivity among Rwandan women living with human immunodeficiency virus.

Response: We have made the change to the title.

Abstract

Here, authors stated that they screened women from 2016 to 2020, but in the paper, but line 102 shows they screened participants from 2016 to 2018. Authors should please state the reasons for the discrepancy.

Response: We have harmonized the enrollment period to be consistent from 2016-2018. Just to clarify, there was follow-up until 2020.

Ethics Statement

Authors should please include the ethical clearance reference number.

Response: Annual ethical review and approval reference numbers have been added (lines 114-116 on page 6).

Material and Methods

Since the study is aimed at identifying key determinants of hrHPV among women living with HIV, it is unclear why the authors did not include women uninfected with HIV as control.

Authors mentioned that the collected data on viral load (line 117) but presented none. Was it an omission?

Response: We did not include HIV-negative women because our original funded proposal targeted women living with HIV. Over 70% of our study population had missing viral load data, we did analyze these data due to limited power to make meaning comparisons. We have added a sentence on that under statistical analyses (lines 168-170 on page 8) and it was also added as a limitation under the Discussion section.

How was the CD4 T cell counted? Authors should state the controls used for the CD4 T cell count and HPV DNA testing.

Response: CD4 cell count data was collected as secondary data with no details about how testing was performed hence not including information about controls. Regarding HPV DNA testing, we indicated under laboratory testing that the Xpert assay was run according to the manufacturer’s instructions, which includes an internal Probe Check Control that the test must pass for the results to be considered valid. Note that we had a small percentage of tests that were invalid because they failed this internal probe check control.

Results

What was the overall prevalence of hrHPV among women diagnosed of

Response: This comment/question seems to be incomplete but if the reviewer meant the prevalence of hrHPV among women with CIN2+, then it was 93.3%.

Discussion

Authors alluded that the lower prevalence of hrHPV in Rwanda is because of ART coverage in Rwanda. Did the authors assess the rate of participants’ compliance to ART regime in their study and the studies they referenced? It is unclear if the authors assess HPV vaccination among the participants as it could impact on the prevalence rate of hrHPV.

Response: We have published data from different time points that indicated that the prevalence of hrHPV among Rwandan WLWH has been decreasing over time with improving access to ART (https://pubmed.ncbi.nlm.nih.gov/32050023/). However, HIV data including ART use was collected as secondary data from medical records with no details about ART adherence hence not including that information on that in this paper. We also did not include information on HPV vaccination mainly because it was not the focus of this study but also, HPV vaccination in Rwanda was launched in 2011 for 9–15-year-old girls and the women included in our study (30–54-year-old women) could not have been vaccinated.

Reviewer #2

1. This study addresses a crucial public health issue: the intersection between HIV infection and cervical cancer risk, particularly through hrHPV infection. The authors tackle this issue with impressive breadth, drawing attention to a vulnerable population in a region where data is urgently needed. But as I read the introduction, I wonder — could the research question be more explicitly stated? While the rationale is strong, the specific hypothesis or aim feels somewhat implied rather than clearly declared.

Response: We appreciate the reviewer for the feedback and for taking their time to review our manuscript. We made the aim clear at the end of the Introduction section.

2. The connection drawn between the WHO’s 90-70-90 targets and the reality of hrHPV testing availability in sub-Saharan Africa is compelling. The authors rightfully highlight the gap between global goals and local capacities. However, I am left wondering: how was Rwanda’s specific readiness for HPV DNA testing evaluated before the 2019 screening program launch? More context on this local implementation process would strengthen the narrative.

Response: Thank you for the feedback. Our understanding is that Rwanda adopted HPV testing based on the WHO strategy for cervical cancer elimination which recommends using a high-performance test (HPV testing) for screening and the program started by piloting HPV testing in some districts coupled with laboratory capacity strengthening. We have added a comment about how the program was launched in the Introduction section (lines 85-86 on page 5).

3. The methodology stands out for its rigor. The combination of HPV DNA testing, VIA, and confirmatory colposcopy with biopsy is robust and aligns with best practices. I appreciate the detail provided on lab testing and the conservative diagnostic algorithm. Still, does the use of pooled hrHPV typing (except for HPV16) limit our ability to identify the specific subtypes driving disease in this population?

Response: We agree with the reviewer that pooled hrHPV typing limits our ability to assess type-specific risk and this was due to the assay we used which uses pooled channels. However, the Xpert assay channels are based on hierarchical risk of the hrHPV types according to carcinogenicity hence providing sufficient information to assess risk in this population. Additionally, we highlighted this issue under limitations (line 286 on page 13).

4. The sample size — over 5,000 women — is remarkable and lends strong credibility to the findings. The demographic spread across all Rwandan provinces increases generalizability. Yet, I’m curious: were there any regional differences in hrHPV prevalence or immune status that could be relevant for tailoring interventions?

Response: Although most women indicated being originally from the City of Kigali (>93%), we did not find any differences in hrHPV prevalence by province of origin as highlighted in Table 1.

5. The data collection methods are thorough, combining nurse-administered questionnaires with EMR extraction. This mixed approach likely improves completeness, but could the reliance on self-reported sexual and reproductive history introduce recall bias? How might this have influenced the associations observed?

Response: We agree with having the possibility of recall bias for self-reported measures/variables and sexual history could also be impacted by social desirability bias. However, given the associations, which continued to be significant even in the logistic regression models, the bias might have been minimal to impact the magnitude of the estimates and the direction of the associations. And there is no way to get sexual behaviors other than through self-report and electronic medical records are not widely available in Rwanda to extract data on reproductive history e.g., number of births. Questionnaire-based data have been the standard in the HPV research field and considered fairly reliable (and women will certainly know how many births they have).

6. The statistical analyses are appropriately complex and well-executed. Stratified analyses by age and CD4 count bring valuable nuance. That said, the decision to exclude women with CIN2+ from the final analysis raises questions. Could this exclusion have masked relevant associations between risk factors and disease progression?

Response: As indicated above, we excluded women with CIN2+ because of collinearity between that outcome and major outcome of this analysis (hrHPV positivity) and a sentence has been added to clarify it (lines 160-163 on page 8). Also, we excluded CIN2+ from the analysis to make our analysis comparable to the other main studies of HPV prevalence and determinants, such as those published from IARC and the NCI. Traditionally, the CIN2+ has been excluded to avoid conflating determinants of HPV infection with those of progression to CIN2+. However, our estimates are unlikely to change significantly as most HPV positives (~90%) do not have concurrent CIN2+.

7. One of the most striking findings is the strong inverse association between CD4 count and hrHPV positivity, particularly for multiple HPV channel infections. This points to immunosuppression as a major driver of viral persistence. But it also prompts a question: could ART duration or viral suppression levels provide further insight into this relationship?

Response: Yes, we agree that if we had data on those HIV-related measures or variables, our analysis could have been richer. However and as indicated, we did not have access to information on ART duration and most women (>70%) had missing viral load results, and we are not confident about the remaining <30% of VL data, hence not including them due to insufficient power to make meaningful comparisons. We have also added a sentence on this under limitations (line 291-293 on page 13).

8. The multivariable model is well-constructed and interpretable, with adjusted odds ratios that clearly illustrate the strength of associations. However, I wonder whether any interaction terms (e.g., between age and CD4 count, or ART use and number of partners) were explored? Such analyses could reveal more complex dynamics at play.

Response: Based on the reviewer’s suggestion, we have explored interaction and we did not find any significant interactions for variables included in the multivariable logistic regression model.

9. The authors report an overall hrHPV prevalence of 25.5%, which is lower than in many other SSA studies. This is attributed to Rwanda’s successful ART program. It’s a hopeful sign, but could selection bias (due to missing or excluded data) partially explain the lower prevalence?

Response: As indicated in the last response to Reviewer 1’s comment, we have published data from different time points that indicated that the prevalence of hrHPV among Rwandan WLWH has been decreasing over time with improving access to ART (https://pubmed.ncbi.nlm.nih.gov/32050023/) hence partly explaining the lower hrHPV prevalence compared to other populations. That being said, our study population might not be representative of the entire country or the SSA region but the proportion of invalid HPV results was too small (~2%) to have impacted the findings.

10. The paper observes that in women aged 50–54, the expected decrease in hrHPV prevalence with higher CD4 counts was not observed. This is acknowledged but left somewhat unexplored. Could this be due to survival bias? Or might other factors like cervical epithelial changes with age play a role?

Response: We agree that there is a possibility of survival bias for this age group and this was discussed with a comment about the small number of those who are hrHPV positive and the possibility of survival bias for older women, please see lines 278-282 on page 13.

11. The findings regarding sexual behavior — such as number of partners and age at first sex — being associated with hrHPV positivity are not surprising, but still significant. I do wonder, though: to what extent are these factors modifiable in this population? How

---

## [Editor Report · Decision Letter 1]

10 Sep 2025

Determinants of cervical high-risk human papillomavirus positivity among Rwandan women living with human immunodeficiency virus

PONE-D-25-13630R1

Dear Dr. Murenzi,

We’re pleased to inform you that your manuscript has been judged scientifically suitable for publication and will be formally accepted for publication once it meets all outstanding technical requirements.

Kind regards,

Patricia Evelyn Fast, MD, Ph.D.

Academic Editor

PLOS ONE

Additional Editor Comments (optional):

Thank you for your careful attention to the reviewers' detailed and insightful comments. There are a few instances in which you provide an excellent answer to the reviewer but do not include that answer in your manuscript. Sometimes, this is justified, for example if there is no clear and satisfying answer to a hypothetical question. However, you may wish to re-read your response letter and consider adding a sentence for a few instances where a reader may be considering the same question as the reviewer.
---

## [Editor Report · Acceptance letter]

PONE-D-25-13630R1

PLOS ONE

Dear Dr. Murenzi,

I'm pleased to inform you that your manuscript has been deemed suitable for publication in PLOS ONE. Congratulations! Your manuscript is now being handed over to our production team.

Kind regards,

on behalf of

Dr. Patricia Evelyn Fast

Academic Editor

PLOS ONE